# Towards the Optimization of eDNA/eRNA Sampling Technologies for Marine Biosecurity Surveillance

Holly A. Bowers [1,*], Xavier Pochon [2,3], Ulla von Ammon [2] , Neil Gemmell [4] , Jo-Ann L. Stanton [4], Gert-Jan Jeunen [4], Craig D. H. Sherman [5] and Anastasija Zaiko [2,3]

1 Moss Landing Marine Laboratories, 8272 Moss Landing Road, Moss Landing, CA 95039, USA
2 Coastal and Freshwater Group, Cawthron Institute, 98 Halifax Street East, Nelson 7010, New Zealand; Xavier.Pochon@cawthron.org.nz (X.P.); Ulla.vonAmmon@cawthron.org.nz (U.v.A.); Anastasija.Zaiko@cawthron.org.nz (A.Z.)
3 Institute of Marine Science, University of Auckland, Private Bag 92019, Auckland 1142, New Zealand
4 Department of Anatomy, University of Otago, P.O. Box 56, Dunedin 9054, New Zealand; neil.gemmell@otago.ac.nz (N.G.); jo.stanton@otago.ac.nz (J.-A.L.S.); gert-jan.jeunen@otago.ac.nz (G.-J.J.)
5 Queenscliff Marine Research Facility, School of Life and Environmental Sciences, Deakin University, Queenscliff 3225, Australia; craig.sherman@deakin.edu.au
* Correspondence: hbowers@mlml.calstate.edu; Tel.: +1-831-771-4138

**Abstract:** The field of eDNA is growing exponentially in response to the need for detecting rare and invasive species for management and conservation decisions. Developing technologies and standard protocols within the biosecurity sector must address myriad challenges associated with marine environments, including salinity, temperature, advective and deposition processes, hydrochemistry and pH, and contaminating agents. These approaches must also provide a robust framework that meets the need for biosecurity management decisions regarding threats to human health, environmental resources, and economic interests, especially in areas with limited clean-laboratory resources and experienced personnel. This contribution aims to facilitate dialogue and innovation within this sector by reviewing current approaches for sample collection, post-sampling capture and concentration of eDNA, preservation, and extraction, all through a biosecurity monitoring lens.

**Keywords:** eDNA; eRNA; marine biosecurity; invasive species

## 1. Introduction

Biological invasions have followed human activities for centuries [1], with cross-regional transfer of non-indigenous species (NIS) having amplified rapidly over the last few decades [2,3]. In the marine realm, this is largely attributed to the massive increase in seaborne trade beginning in the 1950s [4,5], which has served as the major pathway for marine biological invasions [1,6,7]. Continued growth in global maritime traffic and an associated 3- to 20-fold increase in global invasion risk is predicted for the next few decades [8]. Disrupting a potential invasion at the earliest stage of propagation is key, since downstream eradication in highly dynamic marine environments is difficult at best. Although managing the spread of unwanted organisms remains a high priority for regional, national, and international jurisdictions (e.g., Marine Strategy Framework Directive; European Union (EU) Invasive Species Regulation; New Zealand Biosecurity Act), a lack of operational tools and technologies for early detection has been a long-term hurdle [9]. Molecular methods have been used for decades to aid environmental monitoring [10]. The use of tissue or blood samples from an individual to obtain a genetic signal alleviates issues surrounding taxonomic identification, while still relying on visual detection and collection of specimens at the sampled area [11]. More recently, the application of DNA and RNA (collectively termed nucleic acids (NAs)), recovered from environmental samples and referred to as environmental DNA (eDNA) and RNA (eRNA), is increasingly advocated for to be used in biodiversity assessments (e.g., [12–14]). The non-invasive manner of sampling

and non-reliance on visual observations, combined with the advancements of molecular techniques [15,16], has resulted in a rapid expansion of research to determine if this novel monitoring method meets surveillance and management needs in marine (e.g., [17]), freshwater (e.g., [18]), and terrestrial (e.g., [19]) systems. As with any detection system, balance must be struck between precision, accuracy and sensitivity of methods, logistical requirements, and associated costs. These considerations must further be balanced within the larger framework of resource management decisions [20–22].

The sensitivity of NA-based techniques has led to the unprecedented ability to characterize organismal assemblages across multiple trophic levels (e.g., [23,24]), detect and monitor rare species in complex environments (e.g., [25]), expand the known range of organisms (e.g., [26]), and identify potential threats (e.g., [27,28]) that have eluded traditional sampling approaches (e.g., electrofishing, imaging, settlement plates, dive surveys, plankton tows). While these examples demonstrate the immense benefits of using NA-based methodologies for screening environmental samples, several comparison studies have revealed apparent limitations in signal detection [29,30] that can originate from a variety of error sources within the workflow—from sample collection through to processing and bioinformatics (e.g., [31–34]). These limitations can be particularly problematic in trying to meet the demands of high quality assurance/quality control standards in areas such as biosecurity [18,22,35], involving the detection, monitoring and management of pests, diseases, and nuisance species that can cause problems for humans, animals, plants, or the environment (https://www.marinebiosecurity.org.nz/what-are-marine-pests/; accessed on 13 April 2021).

There is an ever-growing body of research aimed at methodological comparisons of NA-based technologies, starting with sample collection techniques through data analyses pipelines. However, no agreement has been reached on best practices for sample collection and processing (e.g., [36] and references therein). Given the explosive growth in the use of eDNA and the emergence of eRNA [37] in myriad environmental contexts, the establishment of a fully standardized protocol throughout the sample collection and processing pipeline is a challenging task, particularly in light of the reporting of inconclusive or contradictory results [35]. Nevertheless, the development of best practice guidelines for each environment that address biosecurity surveillance needs is desirable, especially for the steps encompassing sampling through to the extraction of NAs. Here, we review the challenges for using NAs within a marine environment, and the current methods for sample collection, post-sampling capture and concentration of eDNA, preservation, and extraction through a biosecurity monitoring lens. We end with an outlook on how emerging methodologies can shape the future of invasive species detection. See Table 1 for common definitions, as used in the context of this manuscript.

**Table 1.** Common definitions as used in the context of this manuscript.

| Term | Definition |
|---|---|
| Acid washing | A common practice for decontaminating surfaces and apparatus from organismal material that may contaminate various steps in the pipeline (from sample collection through molecular analyses). |
| Bead-beating | Mechanical way to disrupt cells; filters are placed into a tube containing beads and lysis buffer, and placed on a shaking bead beater for a fixed amount of time. |
| Bioinformatics | Suite of software tools used to analyze genetic data. |
| Capture efficiency | How well a method retains genetic material—for example, material or pore size can affect utility of a filter. |
| cDNA | The DNA strand that is complementary to RNA; part of the intermediate step between genomic DNA and protein, used as a measure of gene activity. |

**Table 1.** *Cont.*

| Term | Definition |
|---|---|
| Concentrate | Through filtration, the contents of a sample are distilled into a smaller volume, thereby increasing the chances of capturing rare or low-abundance organisms. |
| Cross-contamination | When genetic material from target or non-target species contributes inaccurately to molecular analyses, due to inadequate decontamination of surfaces and apparatus. |
| Decontamination | Sterilization of surfaces and apparatus from organismal material that may contaminate various steps in the pipeline (from sample collection through molecular analyses). |
| Degradation | The breaking down of genetic material (DNA/RNA) through enzymatic action (DNA/RNAses) or abiotic factors (e.g., temperature, UV light). |
| Diluted bleach | A common practice (10–50% for >10 min) for decontaminating surfaces and apparatus from organismal material that may contaminate various steps in the pipeline (from sample collection through molecular analyses). |
| Dissolved eDNA/RNA | Free-floating, naked nucleic acid (DNA/RNA) in the water column (i.e., not contained within or adsorbed to any particles). |
| DNA | Deoxyribonucleic acid; central storage of genetic information for organisms (except RNA viruses). In eDNA, analyses targeted for gene presence of species (single or multiple species). |
| DNAse | Deoxyribonuclease; group of enzymes that can degrade DNA, thereby affecting quality and quantity. |
| False positive | An instance where a sample should have been negative, but the result was positive; contamination from improper sample handling can lead to a false positive. |
| False negative | An instance where a sample should have been positive, but the result was negative; inhibitors can lead to a false negative. |
| "Fit-for-purpose" | The concept that a pipeline (sample collection through data analysis) needs to be formulated for each particular sampling context. In contrast to "one-size-fits-all". |
| Inhibitors | A variety of substances of known (e.g., tannins, humics) and unknown type that can be co-extracted with nucleic acids and hinder the performance of downstream enzymatic reactions (e.g., the amplification steps in quantitative polymerase chain reaction (qPCR) and metabarcoding). |
| Metabarcoding | A genetic method that amplifies homologous gene(s) across species in order to gain perspective into the taxonomic constituents of a community. |
| Molecular signal | Results derived from any number of assays (e.g., qPCR, metabarcoding) that detect and possibly quantify genetic material. |
| Niskin bottle | A columnar sampling bottle that can be triggered to capture a whole water sample from a desired depth in the water column. |
| "one-size-fits-all" | The concept that one pipeline (sample collection through data analysis) can be formulated and used in all field and experimental contexts. In contrast to "fit-for-purpose". |

**Table 1.** *Cont.*

| Term | Definition |
|---|---|
| Particle size | Refers to an array of particle types that may be encountered in the water column: whole cells, broken/damaged cell pieces, and naked nucleic acids from lysed cells. Any of these forms can be free-floating or adsorbed to other (non-)organic material. |
| Plankton tow | Vertical or horizontal pull of a specialized net to filter and concentrate water column contents into a smaller volume, thereby increasing the chances of capturing rare or low-abundance organisms. |
| Precipitation | Concentration and purification of nucleic acids (DNA/RNA) through chemical means. |
| Preservation | Near-immediate immobilization of a sample (through a combination of buffers/freezing and transport/storage conditions) to maintain integrity of genetic material. |
| qPCR | Quantitative polymerase chain reaction; also called real-time PCR, because the amplification of a genetic target can be monitored during the reaction, and a determination of copy numbers of that target can be made. This is in contrast to PCR, which cannot be monitored in real time and produces a qualitative (positive/negative) result. |
| RNA | Ribonucleic acid; intermediate step between genomic DNA and protein, used as a measure of gene activity. |
| RNAse | Ribonuclease; group of enzymes that can degrade RNA, thereby affecting quality and quantity. |
| Settlement plates | Artificial structures (e.g., plastic polymer material) deployed in aquatic environments for passive sampling of marine biofouling; can be used to study recruitment of sessile taxa and non-indigenous species surveillance. |
| Snap-freeze | A method to immediately preserve genetic material after sample filtration; the filter is housed in a tube and submerged in liquid nitrogen or on dry ice. |
| Sterility | Maintaining a clean/aseptic environment throughout the entire pipeline to eliminate cross-contamination of organismal genetic material between samples at all stages (sample collection through processing and molecular manipulations). |
| Total eDNA/eRNA | Environmental DNA/RNA in all forms (whole and partial cells, free NAs in solution (dissolved) or adsorbed to particles). |
| Van Dorn | Large chamber water sampler that allows for sampling from one depth or a composite of several depths. |

## 2. Applications of Molecular Tools for Marine Biosecurity Surveillance

Targeted and multispecies molecular techniques are increasingly promoted for marine biosecurity applications [18,38–41], including early (pre- and post-border) detection of unwanted organisms, identification of putative NIS, surveillance of high-priority pest species, determination of the source and pathways of invasion, as well as the genetic structure of founding populations [42]. One advantage of using molecular approaches for surveillance purposes is their sensitivity in detecting and identifying NIS when populations (and therefore concentrations of expelled NAs) are sparse; however, there are numerous considerations to take into account (reviewed extensively in [43]). This aspect, for instance, brings an additional challenge for sampling representativeness, requiring increased field replication [44], sample volumes [45,46], or both, to reduce the risk of false negative results [31,47]. The ability to effectively concentrate NA material over large spa-

tial/temporal scales would provide a valuable trade-off for achieving desirable sensitivity at an economical level, both cost- and timewise (Figure 1).

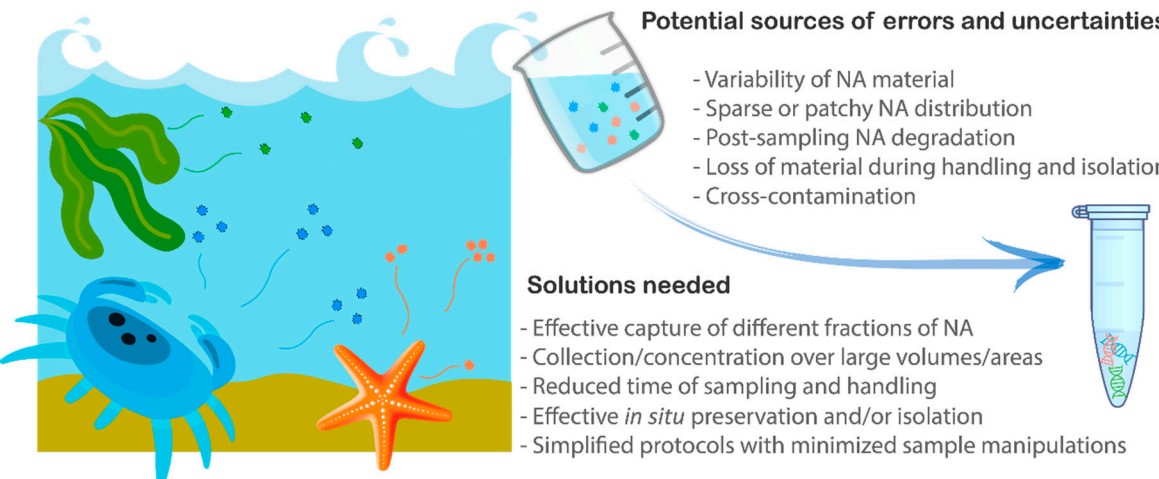

**Figure 1.** Potential sources of detection errors (false negatives or false positives) and uncertainties within the primary steps of the molecular surveillance workflow, from sample collection through to nucleic acid (NA) extraction. (Figure created with BioRender.com).

The genetic material most routinely recovered from an environmental sample is DNA [48], while RNA is used to provide additional insights into the ecology of alive and viable organisms within biosecurity contexts [49–53]. The stability of DNA, which can persist for extended periods in the environment, depending on the conditions (days to years [54–56]), enables easier handling of samples until extraction. However, the influence of varying abiotic factors on the persistence of eDNA complicates the sampling design and interpretation of results, e.g., in the context of the current presence and affinity of a viable population of target (pest) species [57–59]. On the other hand, RNA is a fragile molecule that degrades rapidly (typically hours to days [12]). The fast degradation of RNA allows better approximation of the signals from alive and viable organisms in a sample [49]. The fate and prevalence of particle-bound eRNA is currently not well understood, questioning the applicability of eRNA for routine monitoring purposes [12].

The variable states that NAs are found in the environment, ranging from free-floating molecules to molecules bound to larger complexes (inorganic particles, cell debris, organic floc, tissues, scales), or even intact organisms and their propagules, introduces another level of complexity for effective NA capture from a sample [44]. The decisions to be made at this step (e.g., optimal filter type, pore size, need for pre-filtration or precipitation) are often impeded by poor knowledge of target NA fractions and their variability (spatial and temporal) in the environment. A variety of factors within the marine context can influence this variability and detectability: salinity, temperature, advective and deposition processes, hydrochemistry and pH, and contaminating agents (e.g., [60–63]). Inherent biological factors, such as cellular degradation (influenced by cell health) can also affect detectability [62]. All of these influences need to be taken into consideration when optimizing protocols around "sampling windows" as part of any sampling approach, from grab samples to passive sampling devices.

Another challenging aspect for biosecurity practitioners and the broader application of molecular techniques is related to the complexity of sample collection and handling protocols [21,22,64]. Thus, the exceptional sensitivity of molecular detection requires an abundance of caution to prevent cross-contamination at every stage of the workflow and avoid false positive detections—the major concern of biosecurity managers hesitant to implement molecular surveillance [18,35]. Furthermore, the nature of the sampled material makes it prone to degradation (e.g., [65]). This can be prevented by minimizing handling

time in the field and stabilizing the material as soon as possible after collection, in a cost-efficient and preferably non-toxic way (Figure 1).

In the following sections, we review the current methods available for (i) sample collection, (ii) post-sampling capture and concentration of NAs (including filter types), (iii) preservation, and (iv) extraction and subsequent storage, all within the utility of the marine biosecurity surveillance sector. The major existing gaps are addressed in the context of future perspectives. We do not consider sampling design in this review, as it is case-specific and difficult to generalize. A fit-for-purpose sampling strategy should ultimately be developed for each coastal location, taking into account parameters like harbor size, local hydrodynamic peculiarities (exposure, tidal regime), proximity to introduction pathways, potential sources of inhibitory contaminants, and historical detection of target species.

### 3. Sample Collection

Testing for the presence of invasive species via NAs within the biosecurity sector ideally requires inexpensive technology that is easy to use across all skill levels, is not prone to cross-contamination, and consistently leads to robust and reliable results. Identifying equipment that meets these criteria will help with its adoption into sustained monitoring frameworks. This rapidly growing field has predominantly incorporated existing methods for sample collection (e.g., [66]); however, there have been calls from management entities for fit-for-purpose technology (e.g., [67]), especially since many traditional methods pose constraints for eDNA sampling [21,43].

Marine biosecurity surveillance targets a whole range of organisms representing different taxonomic groups, functional traits, and lifestyles—from sediment-dwelling or biofouling invertebrates to algae and fish inhabiting the water column. Also, in the case of untargeted surveillance (e.g., passive "screening" for new incursions), the life form, behavior, abundance, and peculiarities of NA distribution of the organisms are generally unknown. Therefore, it is difficult to develop a sampling protocol that is generalized across the multiple habitats in which an incursion may occur. Sampling different environmental matrices, such as seafloor sediments, biofouling of infrastructure, and the water column (which is a rather complex environment in itself), may not always be possible due to logistical constrains (e.g., accessibility of sampling sites in busy ports), health and safety risks (e.g., sample collection by SCUBA divers from open coast hard bottoms), and associated cost and time efforts (Table 2). A few studies have compared the efficiency of targeting different matrices for biodiversity assessment and species detection [66,68–70]. Some studies have found important differences between, e.g., sediments and water at different depths [71–73]. A comprehensive biosecurity surveillance program should ideally consider collecting representative samples from different habitat types, enabling detection of a wide range of organisms (e.g., [74]). However, in the context of often limited and consistently diminishing monitoring resources, it is important to find a meaningful trade-off between cost-efficiency and the extent of surveillance effort ([75] and references therein). Surface water samples are usually logistically easier to collect and are standardizable at larger scales [76]; thus, they can be considered an efficient alternative for obtaining signals from both planktonic and benthic communities [75,77]. Hereinafter, we focus on methods relevant for waterborne NAs (unless stated otherwise).

**Table 2.** Considerations in the nucleic acid (NA) surveillance workflow with regards to choice of biological matrix to sample, as well as possible approaches to collection, concentration, and preservation.

Field Workflow Considerations column headers:
- Large Volume/Area Coverage
- Lower Inhibitor Effect
- Possibility of Visual Pre-Screening of Biodiversity
- Selectivity for Invasive Taxa
- Affinity of Signal to Source Location
- Homogeneity of Material (Effective Replication)
- Non-Disruptive to Ecosystem
- Non-Hazardous (H&S wise)
- Capture of all NA Fractions
- Suitable for Varying Depth/Locations

Lab Workflow Considerations column headers:
- Easy/Inexpensive
- No Specialized Equipment/Infrastructure Required
- Possible in Field
- Lower Risk of Compromising Integrity of NAs
- Time-Efficient
- Low-Waste
- Non-Hazardous
- Efficient Cross-Contamination Control

Field Workflow Considerations rows:
- **Soft sediment**
  - Benthic samplers
  - Scuba diving
- **Hard bottom**
  - Scuba diving
  - Settlement plates
  - Biofouling on artificial structures
  - Scuba diving
  - Settlement plates
- **Water column**
  - Whole water samples
  - In-situ filtration
  - Plankton nets

Lab Workflow Considerations rows:
- **Post-sampling concentration (water samples)**
  - Centrifugation
  - Precipitation (chemical)
  - Filtration
- **Preservation**
  - Snap-freezing
  - Preservation buffers
  - Desiccation

**Color key**
- Green: Yes
- Orange: No
- Blue: Neutral/context dependent

The majority of studies targeting waterborne NAs have employed sampling with bottles, carboys, or buckets (or some combination), and have been amended with a filtration step for sample concentration when needed [58,78]. Plankton net tows have been used for concentrating material from a large volume of water [30,42,53,70,79], an approach that is desirable for detecting low concentrations and overcoming the patchiness of targets (e.g., [80]). However, the configuration of these nets renders them highly prone to cross-contamination across sampling sites/dates, given the propensity of cells and particles to stick in the numerous crevices. Furthermore, the nylon used in constructing plankton nets is often not conducive to repeated, bleach-based sterilization (von Ammon, Pochon, and Zaiko, personal observations). Alternatively, the plastics used for Niskin bottles and van Dorn samplers (both utilized for capturing whole water from discrete depths) are better suited for sterilization via diluted bleach (e.g., [81]) or acid washing (e.g., [82]); however neither of these devices allow for in situ concentration of particles.

The use of sterile disposable bottles can be economically feasible depending on sample size; however, this is not an option for bulk water collection devices and downstream filtration apparatus. More commonly, sampling containers and equipment are rigorously sterilized with bleach [83–85] or acid [82,86,87] and rinsed several times with deionized

water, followed by site-specific water to remove sterilizing agents. False positives can occur with any breach in the protocol (e.g., reduced sterilization times, improper concentrations). Nucleic acid/contaminant removal is of the utmost importance for NA sampling; however, the above measures can be cumbersome in a biosecurity context, where considerable time, effort, and oversight are needed to ensure proper decontamination steps are carried out.

Recent gear adaptations provide promising alternatives for reducing sample handling and associated contamination. Adrian-Kalchhauser and Burkhardt-Holm [64] modified a commercially available telescoping water sampler to combine the capability of discrete depth sampling with the ability to use a new bottle for each deployment. This technique still requires decontamination of the main apparatus between samples. However, the water mass is collected in a separate disposable vessel, and is not subjected to a re-useable chamber (e.g., van Dorn or Niskin bottle). The Smith–Root eDNA backpack sampler [88] is a "fit-for-purpose" system for sampling waterborne eDNA. The unit monitors and regulates the pressure and flow of water across a collection filter. Partially biodegradable filtration units can be used to self-preserve the filter via desiccation, and this has been shown to be comparable to ethanol preservation, even over long storage periods [88]. The use of a negative-pressure inline filtration system means that the water sample can be taken up directly by the intake tubing to the filter, while all pumping occurs downstream of the filtering unit. This approach drastically reduces the chance for cross-contamination between sampling events. Newly configured autonomous platforms are in development and offer great promise towards more comprehensive sampling approaches (Mesobot, [89]; 3G Environmental Sample Processor (ESP), [90]).

One prominent disadvantage of many environmental NA sampling systems and strategies is that they offer only point measures for species detection; thus, there is a move towards systems that might aggregate an eDNA signal over time to provide a window into biosecurity threats across a wider time span. One such device is the continuous, low-level aquatic monitoring (Continuous Low-Level Aquatic Monitoring, C.L.A.M.; https://aqualytical.com/, accessed on 13 April 2021) system that allows for a time-integrated sample (up to 36 h), with the option to tow or allow the device to drift to also capture spatial coverage. The device employs filtration and media sequestering steps for sampling totals and dissolved eDNA. Other lower-tech, cost-effective, and environmentally friendly aggregate strategies are also emerging. One such approach is to deploy passive eDNA samplers that capture eDNA to a physical matrix, such as montmorillonite clay [91], artificial reef structures [92], or settlement plate arrays [24,93]. Such strategies could lead to longer-term, low-cost surveillance of eDNA in many systems, perhaps with a reduction in the need for laborious filtering (see below) and a decrease in the use of disposable plastics. Unfortunately, due to its instability, the collection of eRNA is likely to always require sterile, specialized equipment.

Collectively, these new technologies are addressing many of the criteria needed for sampling waterborne NAs within a biosecurity framework. Easy-to-operate enclosed systems greatly minimize the opportunity for cross-contamination between samples. Furthermore, immediate in situ fixation steps eliminate any lag times to sample preservation (described below). Taken together, these aspects help streamline areas in the pipeline where false positive or false negative information can originate. Gaps in knowledge for these types of collection systems include: (1) robustness and reproducibility of different filter types and fixatives across diverse taxonomic groups and varied aquatic environments (e.g., turbid waters, hydrological influences); (2) utility in saltwater environments; and (3) incorporation of pre-filtration steps to allow for greater volumes to be processed, in order to increase detectability of rare taxa.

## 4. Post-Sampling Capture and Concentration of eDNA

Most commonly, waterborne NAs from environmental samples are concentrated via filtration. Filter types and pore sizes have been tested for a variety of target organisms (e.g., [82,94–100]). However, unsurprisingly no "one-size-fits-all" consensus has

emerged. Environmental DNA studies have reported using filters ranging in pore size from 0.22 microns [57,85,101] to as large as 20.00 microns [102], although many publications state pore sizes of one micron or less [94,103]. Ideally, the smallest pore size would be used in order to capture all sub-cellular particles. However, capturing adequate material (particle size and abundance) to maximize meaningful molecular signals downstream must be balanced with the time it takes to filter large volumes of water and the increased chance of clogging, which reduces the volumes that can be filtered. Too much material can potentially introduce inhibitory substances (e.g., humic compounds, suspended sediment) or potentially swamp out the molecular signal from rare targets [104,105]. A limited number of studies have shown that larger pore filters can be as informative as smaller pore filters in detecting even rare targets [106,107]. Results from these studies are promising; however, it is unclear how they will translate across a broad range of trophic levels, particle sizes, and environmental variability.

More data are needed on the amount and type of NA material captured and lost when using various pore sizes, to select the optimal combination of time and signal detection efficiency for particular biosecurity applications. As we move towards increasing that body of data, one option is to filter sequentially through multiple pore sizes. This approach avoids the impact of unpredictable episodic disturbances (e.g., influx of particles after a storm surge; highly dynamic marine and estuarine systems; seasonal variability in allochthonous material) on a standardized sampling pipeline that is limited to one pore size. This strategy also increases the opportunity for capturing a broad range of taxa from different NA fractions, and avoids rapid clogging of the small pore size filter (e.g., [108,109]).

Some studies have employed sequential filtration to assess the optimum pore size for capturing eDNA, and in some cases the remaining effluent harbored a significant amount of eDNA. For common carp (*Cyprinus carpio*), Turner et al. [94] determined that the molecular signal (via qPCR) was most abundant from fractions captured between 1 and 10 microns. However, they also confirmed a large pool of predominately non-target eDNA in their ethanol-precipitated (an alternative to filtration), 0.2 micron filtrate, a fraction that would have remained unexploited in a broad-scale taxa study or monitoring program. Moushomi et al. [110] used qPCR to demonstrate greater copy numbers of nuclear and mitochondrial targets of *Daphnia magna* (a small, planktonic freshwater crustacean) in eDNA that was ethanol-precipitated and extracted from the 0.2 micron filtrate, versus that extracted from the filter itself. Notably, the eDNA decay rate was lower in the effluent compared to material in the <1.0 to 0.2 micron fraction. These studies suggest that subcellular material can be a significant and more stable source of information, and should be explored in future studies.

As an alternative to filtration, Turner et al. [94] and Moushami et al. [110] incorporated ethanol precipitation steps to retrieve eDNA from effluent samples, but this step is limited to small volumes, and can be problematic for seawater samples, due to inherent salt interference (discussed below). However, Sassoubre et al. [86] were able to successfully precipitate eDNA from mesocosm tanks housing two species of marine fishes, and they demonstrated significantly greater genetic copy numbers (via qPCR) for both species in unfiltered tank water (55% and 26%) compared to the cumulative values from three size fractionations (10.0, 1.0, and 0.2 micron filters), plus the remaining effluent. These results warrant additional study to test the reproducibility of this approach for NA-based monitoring.

The majority of studies on environmental NA states and particle sizes to date have been carried out in freshwater environments or have involved a limited number of target species. Extrapolation to a biosecurity framework is complex, given the desire to construct a robust pipeline that maintains an ability to detect newly introduced species, which may include a broad array of taxa that shed particles of different size (e.g., subcellular, colonial), type (e.g., epithelial, slime, waste products), and abundance, depending on organism dimensions, behavior (including seasonality), or physiological state (e.g., egg and veliger release, aging population). In addition, it remains unclear how long the various particle

types stay in suspension, which will affect surveillance decisions. Perhaps casting a wider net at this point in time (i.e., analysis of multiple filtered and precipitated fractions) will limit the potential of missing key species, while simultaneously working towards the development of comprehensive databases and new technologies.

In terms of filter type, cellulose-nitrate filters are most commonly used in eDNA studies, and have been shown to consistently perform well for eDNA capture and extraction [84,111]. However, the current body of literature for filter performance does not yet adequately address areas of interest for the biosecurity realm. It should also be taken into consideration that not all filters are available in all desired pore sizes, with some having regionally limited access or imposed additional costs (e.g., the dangerous goods tax on importing cellulose nitrate filters; authors' personal observation).

To optimize the filtration step for biosecurity applications, available filter types need to be tested for their efficiency in capturing a wide range of NA fractions in waters with varying levels of organic matter and fluctuating water quality parameters (e.g., pH, salinity). Ultimately, we need to balance capture efficiency (range of NA fractions in the presence of organic substances), assess compatibility with eRNA applications, and determine the feasibility of filters in an enclosed engineered system for reduced cross-contamination, thereby diminishing the risks of false positive signals that are highly undesirable and costly for managers and industry.

## 5. Preservation

Rapid preservation of NAs, particularly eRNA, during field sampling is critical. The method used ideally needs to exhibit robust performance across a variety of sample characteristics, including low-to-high biomass, a broad pH range, and the presence of inhibitors. Among current studies, cold storage is commonly used to minimize any loss of genetic information between field sampling and lab processing [112–116]. However, the use of ice, dry ice, or liquid nitrogen for snap-freezing, or cooling facilities requiring electricity and specialized equipment, all present practical limitations in the field and for transportation. Preservation solutions can be better suited, and preferably should be simple, cost-effective, sterile, and without requirements for special permits (i.e., non-toxic).

A cheap, room-temperature preservation option for whole water is isopropanol [117] or ethanol [97,118–120]. Yamanaka et al. [121] found that the preservation of whole water with benzalkonium chloride was successful in recovering 92% of eDNA (up to 8 h). While cheap and effective, these chemicals require the transport of large volumes, which can be a limiting factor when sampling at sea or remote coastal locations. More commonly, samples are preserved after concentration via filtration, using 20% DMSO buffer [122], RNAlater (Qiagen, Germantown, MD, USA), Longmire's buffer [123], or cetrimonium bromide (CTAB; Sigma-Aldrich, St. Louis, MO, USA) buffer [95]. Ammonium salts, dimethyl sulfoxide (DMSO), and RNAlater showed high precipitation, leading to inhibition of real-time PCR assays (qPCR), while CTAB and Longmire's successfully preserved filtered eDNA at 20 °C over a two-week period [95]. Salt-saturated DMSO (DESS) buffers containing EDTA have also been suggested as a preservative [124]. Some preservation solutions have been shown to be biased toward taxonomic groups [125] or species [126], while Longmire's buffer even appeared to enhance eDNA recovery, likely by increasing cell lysis efficiency [127,128]. This could suggest Longmire's buffer as the best option for sample preservation at ambient temperature conditions.

A few studies indicate that snap-freezing of whole water samples instantly after collection outperforms other preservation methods [125,126]. This is true for one-off analyses, but where samples may be used repeatedly, the issue of freeze–thaw-induced degradation becomes important. In our own experience, even one cycle of freeze–thawing can reduce the eDNA signal. The effect of multiple defrosting episodes on eRNA is largely unknown, but is expected to be substantial as well. Therefore, where possible, it is preferable to avoid freezing samples in the field and instead keep them chilled if they can be delivered to the lab within a reasonably short time (up to ~24 h [119,129]), where

they can be immediately processed for filtration and NA extraction (or storage). When specifically targeting eRNA, samples need to be filtered expeditiously and snap-frozen, or stored in expensive RNA-compatible preservation buffers (e.g., RNAlater, LifeGuard, DNA/RNA-Shield, etc.) in the field. Quaternary ammonium salts have been reported to be efficient preservation buffers in some [130,131] but not all [132] RNA studies.

Recent technologies have explored desiccating, self-preserving, and fully encapsulated filter membranes (e.g., Smith-Root self-preserving eDNA Filter Packs), which could achieve higher detection sensitivity on field samples than with buffer preservation [88]. Sterivex and Acrodisc syringe filters work with similar technologies using silica desiccant. Syringes are then chilled at 0 °C and stored at −10 °C or lower until extraction. There is, however, a large knowledge gap in how these approaches perform for eRNA.

## 6. Extraction of Nucleic Acids

Extraction of high-quality NAs from environmental samples is paramount for successful, molecular-based marine biosecurity surveillance. A plethora of NA extraction methods have been developed since the inception of DNA isolation in 1869 [133]. The two fundamental steps for extracting NAs are (1) cell lysis and extraction of intracellular NAs into aqueous solution—this is done either enzymatically by incubation with hydrolyzing enzymes (e.g., proteinase K; Sigma-Aldrich, St. Louis, MO, USA [134,135]); mechanically through bead-beating, freeze-thawing or grinding [136,137]; or chemically, using detergents (e.g., sodium dodecyl sulfate (SDS; Sigma-Aldrich, St. Louis, MO, USA), guanidinium thiocyanate (Sigma-Aldrich, St. Louis, MO, USA), or CTAB (Sigma-Aldrich, St. Louis, MO, USA)) that solubilize cell membrane components [138–140]—and (2) purification and isolation of NAs from the aqueous phase. This is done via washing with detergent and/or organic solutions (e.g., phenol–chloroform–isoamyl (PCI) [141]), precipitation with isopropanol, ethanol, or polyethylene glycol [142,143], as well as filtration through gels, silica columns, magnetic beads, or ion-exchange resins [142,143].

While traditional lab-based NA isolation protocols using SDS, CTAB, or PCI are the least expensive, and have been successfully applied for monitoring aquatic biodiversity from both pelagic and benthic aquatic environments [94,138,144–146], they are time-consuming and include the use of toxic chemicals that restrict their application to specialized personnel and labs only [36]. Additionally, traditional lab-based protocols are often modified by laboratories to improve the characterization of either specific groups of organisms or sample types, and therefore no single such protocol will ever fit all the situations required by marine biosecurity surveillance programs. Commercial kits are now most commonly used in eDNA studies [36,124]. They present significant advantages over traditional lab-based protocols, including ready-to-use QC/QAed reagents, standardized extraction procedures, and PCR inhibitor removal solutions, with an absence of dangerous chemicals and relatively low cost and time per sample. In the biosecurity context, standardization is critical [35], and therefore the use of NA extraction kits is highly desirable, as they ensure an appreciable degree of consistency, which can be optimized through the automation of extraction using robotics [147]. Despite these advantages, no universal extraction kit that performs best for all sample types and research goals has been developed; there exist numerous DNA or RNA kits designed for specific applications, but a comparatively much smaller selection of kits enabling the co-extraction of both DNA and RNA from environmental samples [145].

Lear et al. [148] undertook a comprehensive data mining of 584 research articles, focusing on "extracellular DNA", and recommended the use of specific kits for DNA extraction of organisms in environmental samples, as follows: (1) Qiagen DNeasy PowerSoil and/or PowerMax kits from soil, sediment, feces and leaf litter; and (2) Qiagen DNeasy Blood and Tissue kits and/or DNeasy PowerWater kits for extraction of DNA from water and ice. Other researchers have also compared a number of commercial kits, with the leading contenders being Qiagen DNeasy Blood and Tissue and the PowerWater kits (see [149] for review). Hinlo et al. [150] found the DNeasy kit gave better DNA yields, however

PowerWater has an additional step to remove downstream enzyme inhibitors. This means that even though the nucleic acid yield may be higher for the Qiagen DNeasy Blood and Tissue kit, for samples high in naturally-occurring PCR inhibitors, PowerWater may give greater sensitivity and accuracy. Therefore, judicious selection of a commercial kit best suited to the environmental sample can significantly improve outcomes. Comparable estimates of biodiversity have been described from these two kits; however, representation should be investigated prior to committing to a strategy [29,82].

A significant challenge for environmental NA studies arises from the co-purification of naturally occurring enzyme inhibitors. The most well-described of these are humic, tannic, and fluvic acids [151–154]. At high concentrations, these inhibit polymerases, preventing downstream detection of target sequences and potentially interfering with metabarcoding studies. Two approaches are generally employed to prevent inhibitory molecules from biasing environmental nucleic acid surveys: using nucleic acid extraction systems that remove inhibitors (e.g., PowerWater or other inhibitor removal kits, [150,155]), or the use of downstream polymerase enzymes that are resistant to inhibition. A new technique built around hydroxyl-coated magnetic beads offers the promise of rapid, inhibitor-free, high-yield nucleic acid isolation from environmental samples. In a pioneering study undertaken by Yuan et al. [156], this method increased yield from approximately 10% for traditional methods to over 90% from wastewater treatment plant-activated sludge when magnetic beads were used. Yuan et al. [156] was able to fine-tune the eDNA fraction captured using magnetic beads by combining with sample filtration to examine free DNA, intracellular DNA, or DNA absorbed to extracellular particles. Sanches and Schrier [157] combined glass-fiber filters and magnetic beads to capture eDNA from estuarine samples that they argue contain some of the highest levels of inhibitory molecules for any environment. However, with only these studies to date, further work on volume limits, ratio of magnetic beads to sample, and nucleic acid purity all require further investigation.

For marine biosecurity applications, a fundamental problem with available kits, apart from surety of supply, relates to the actual amount of biofouling or sediment material that can be processed, ranging from 0.25 g (DNeasy PowerSoil; Qiagen, Germantown, MD, USA) to 10 g (DNeasy PowerMax; Qiagen, Germantown, MD, USA), with the latter being approximately four times more expensive than the former. Therefore, biosecurity researchers often face the difficult choice of either increasing the replication of homogenized (e.g., freeze-dried or bead-beaten) samples; extracting DNA from only 0.2 g, using the cheaper kit unless kits can be modified (e.g., [83]); or processing fewer samples while maximizing the recovery of rare species from larger amounts of starting material, using the more expensive kit. A combination of approaches may be employed, which increases the difficulty of performing statistical analyses.

Another significant problem for long-term biosecurity surveys is the proprietary nature of reagent recipes, as well as the systematic fine-tuning of available commercial kits that offer no guarantee for extended consistency in protocols and results. The recent discontinuation of the former DNeasy PowerSoil kit (now PowerSoil Pro; Qiagen, Germantown, MD, USA) is a good example [147]. Nevertheless, while non-commercial protocols can be easier to maintain through time, and provide higher amounts of extracted material, they are much more likely to introduce contamination in the workflow and generally yield much lower NA quality compared with commercial kits, which are able to more effectively remove PCR inhibitors. Therefore, one could argue that quality is more important than quantity for minimizing false positive/negative results, and that the use of commercial kits with appropriate replication and sample isolation workflow is the best approach for routine marine biosecurity surveillance efforts. Integration of an internal positive control (IPC) into any extraction protocol (to assess extraction efficiency and impacts from inhibitors) needs to be considered as part of a routine pipeline to ensure a robust downstream dataset (e.g., [158,159]).

Methods for isolating eRNA are broadly similar to those targeting eDNA; however, the generally acknowledged (though recently challenged) view of increased eRNA susceptibil-

ity to degradation makes this area more challenging [12,50,160]. eRNA recovery rates have been reported to range from 70% to as little as 5% of the original concentration [161,162]. This is reportedly due to the rapid degradation that can occur during standard enzymatic treatment with DNAse (to avoid cross-contamination from carry-over genomic DNA) and the reverse transcription process (used to generate cDNA), both of which involve long exposures to the elevated temperatures needed for enzyme activation/de-activation. Without actively inhibiting or excluding environmental RNAses through using standard RNA handling precautions, reliable and reproducible capture of eRNA will be difficult and inconsistent.

In the current era of rapidly evolving molecular and analytical technologies, it is particularly important to build robust and quality-assured baselines (e.g., in the form of eDNA/eRNA sample and data archives) for ensuring continuity and intercalibration of ever-emerging tools [163]. Therefore, it is important to ensure that extracted NAs remain stable, and there is no or minimal degradation over time. Post-extraction stabilization of NAs can be achieved through deep-freezing or appropriate buffering. Ideally, curated cryopreservation archives should be established for long-term sample storage and back-ups. Moreover, access to such archives would also allow retrospective refinement of biodiversity information if required for conservation or biosecurity purposes.

## 7. Outlook

Several factors that can introduce variation and uncertainties when sampling NA within a marine biosecurity context still need to be adequately addressed [47,163], in order to ensure that the produced data meet the requirements for precision and accuracy, and are balanced with logistical and economical constraints. First, the effects of salinity, advective and deposition processes, peculiar hydrochemistry, and contaminating agents may compromise the capture of NAs. Second, unique workflow requirements are needed for optimized eRNA-based surveillance, which has recently been advocated as a promising alternative for biodiversity science [50], and more specifically, for biosecurity surveillance [12,49,69]. Third, there is a need for simplistic, yet robust and sensitive, non-toxic collection procedures (e.g., minimal sample volumes, enclosed filtration and subsequent elution and extraction of material, passive sampling options) that do not compromise sample integrity (e.g., through degradation or contamination) in the hands of non-laboratory personnel. In a marine biosecurity context, optimized fit-for-purpose and time-efficient technologies for sampling and purifying NAs from environmental samples would transform our ability to undertake cost-effective surveillance, aid in early detection of potential threats, and revolutionize biosecurity management potential (Figure 2). Refinement and validation of NA-based methods to fit these criteria will enhance molecular surveillance by biosecurity practitioners and citizen science programs.

In order to address the outstanding challenges of molecular (NA-based) biosecurity surveillance, we should proactively explore and embrace methodological advances in environmental genomics and those technologies emerging in adjacent fields (e.g., chemical, biotechnological, or medical applications). For example, nanofibers and nanoparticles used for immobilizing and the delivery of DNA [164–166] can be explored for applicability in eDNA/eRNA sampling devices. Selective binding and stabilization of waterborne NAs would be beneficial for both snapshot and time-integrated sample collection, with the latter of particular interest for biosecurity applications, as it would allow for increased detection probabilities of weak molecular signals from new incursions in highly dynamic coastal environments. The concept of time-integrated (or "passive") sample collection is well-exploited for cost-effective monitoring of nutrients and contaminants in water [167,168]. However, such technology has not yet been extended to NA capture. There is a range of common compounds with promising properties for NA adsorption from ambient water, including clay minerals, silica, and alumina [91,169–172], as well as less conventional substrates that show potential for NA binding, e.g., functionalized graphene, gold nanoparticles, polyamidoamines [165,173,174]. The use of marine invertebrates, such as sponges [175], as

natural samplers and bioaccumulators, is also an area of promise for ongoing, low-cost, surveillance, and is worthy of being explored further.

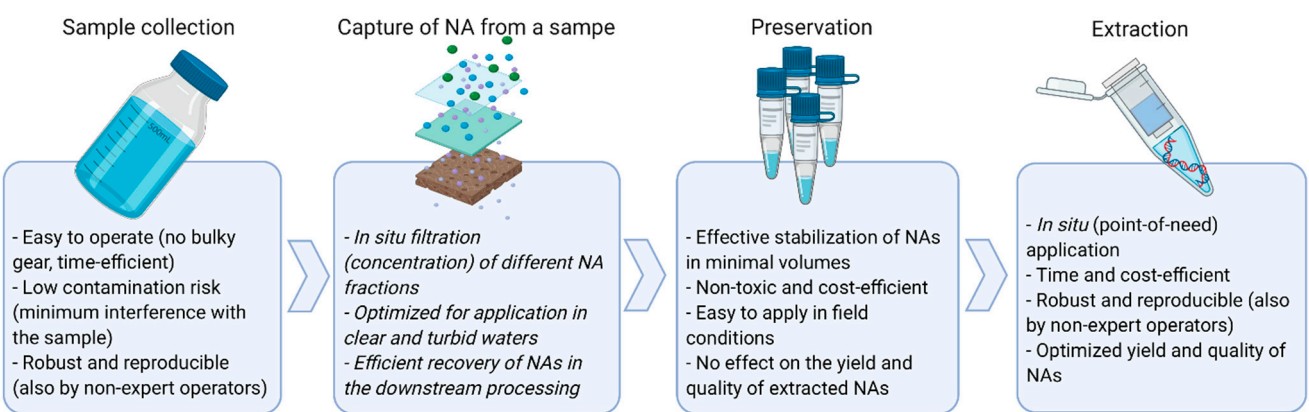

**Figure 2.** Desirable features of a methodological workflow around eDNA/eRNA sample collection through to nucleic acid extraction for robust implementation in routine biosecurity surveillance (Figure created with BioRender.com).

Recent advances in the field of microfluidics has resulted in the development of several approaches for separating, sorting, concentrating, and characterizing microparticles [176]. Non-membrane-based approaches are finding application in biomedical, industrial, and environmental fields, and allow for the separation of particles with particular characteristics [177–179]. These approaches use the motion of suspended particles in fluids to concentrate and separate them out based on size, shape, and density (inertial microfluidic devices), allowing for much larger volumes of liquids to be processed, but require information about target particle characteristics (e.g., size, shape, density). For most marine pests, the size/density characteristics of the suspended material harboring NAs remains unclear and needs to be better characterized. Nevertheless, these new technologies offer the potential for greatly increasing water sample volumes and NA capture, thereby significantly increasing our chances of detecting low-abundance species.

We also see value in developing sequence-specific hybridization of NAs, based on substrates enriched with custom synthesized biomolecules (e.g., peptide nucleic acids) to bind with high specificity to NA strands originating from a target species (e.g., NIS or pathogen). Peptide nucleic acids are synthetic polypeptide backbones with nucleic acid bases attached as side chains, that can form Watson–Crick pairings with complementary DNA. The artificial, synthetic, polyamide backbone increases resistance to degradation by nuclease and protease enzymes in the environment [180]. Protein−nucleic acid interactions are an important field of modern molecular biology, and have been extensively explored for developing DNA machineries and biomedical applications [181,182]. However, there are significant hurdles to overcome to identify materials suitable for highly dynamic biological and physico-chemical aquatic environments. There is a requirement for high specificity for target organism(s), and resilience to changes in environmental conditions and non-organic contamination, as well as the potential to recover genetic material for downstream analyses (NA extraction and quantification).

Targeted eDNA capture can be achieved via DNA enrichment techniques, using methods recently developed to remove non-target DNA from a DNA mixture in order to increase the probability of detecting the target [183]. DNA complementary to the targets is synthesized with biotin attached, and then bound to streptavidin-labelled magnetic beads that can be held by magnets. The DNA capture beads are commonly known as "baits". Baits were recently used to isolate low concentrations of ancient Pleistocene human DNA from soil samples collected from cave floors [184], and DNA target enrichment has been used to increase the DNA barcoding regions for the metagenomic characterization of aquatic invertebrate communities and remove non-target organisms, such as bacteria [183].

The performance of these approaches for environmental applications, however, should be thoroughly examined before they can be considered for biosecurity surveillance. For example, it is important to assess their specificity for different fractions of environmental NAs (versus binding non-target compounds); sensitivity or ability to absorb NAs at low concentrations; cumulative capacity—that is, the ability to accumulate NAs over a sustained period of time; saturation and magnitude of NA signal/integrity loss over time; and compatibility with common NA isolation procedures and analytical workflows. It is also crucial to carefully evaluate the cost-efficiency of these techniques, as it might determine the attractiveness of their uptake for routine biosecurity applications.

As biosecurity depends on finding rare NA signals, and those from biologically viable taxa, there is a strong need for more improvements in the eRNA sampling pipeline. To that end, challenges unique to eRNA need to be taken into consideration throughout the myriad steps that cross-cut aspects of collection, concentration, and preservation. The increase in time and financial resources need to be weighed against the benefits of the information that can be gleaned from eRNA. Furthermore, it would be advantageous for eRNA-specific characteristics to be tested and included in the design of deployable systems that are easy to use (few manipulations) by all skill levels, yet remain reliable and not conducive to contamination or impacts from environmental variables. With increasing efficiency in laboratory processing, eRNA detection limits can improve and are promising to deliver sensitive information about the living fraction of the sampled biomass.

Recent technological advances now mean that the gap between sample collection and laboratory-based diagnostics is narrowing, with many assays now looking to circumvent or simplify the extraction step. Such advances are often driven by the development of new in situ diagnostic approaches that offer rapid, user-friendly, sensitive, and cost-effective field-ready tools (point-of-care or point-of-need tests). Among the most promising approaches are loop-mediated isothermal amplification (LAMP [185]) and recombinase polymerase amplification (RPA), both of which involve the rapid amplification of DNA/RNA using an isothermal amplification reaction with similar sensitivities and specificities to laboratory-based PCR assays [186,187], as well as the CRISPR/Cas13-based SHERLOCK system [188,189]. These approaches do not require thermal cycling and operate at much lower temperatures, with the end-point analysis of amplified products in the field possible using lateral flow (LF) strips, similar to those used in pregnancy test kits [190]. These assays have already been developed and applied to a number of pests, diseases, and ecological monitoring [187,188,191,192], but have not yet been widely applied in the marine biosecurity context. The assays are particularly useful for rapid field-based diagnostics, since once established, the tests themselves require minimal technical skills and laboratory infrastructure. However, extraction of high-quality nucleic acids is still needed, and represents a bottleneck in the widespread adoption of RPA and LAMP for point-of-care tests [193]. Several new, simple, and rapid extraction methods have been developed [194–196], but have not been widely tested for environmental applications.

The emergence of simple, low-cost biosecurity tools also opens up the prospect of engaging the wider community and citizen science groups to undertake monitoring and surveillance [9]. A time when NA-based surveillance data might be collected by school children, community groups, and local agencies directly using devices that produce geospatially tagged genotype or sequence data in the field is becoming a reality. Currently, devices such as the Nanopore Flongle [197,198] provide such capability, and this will only become more portable and integrated with our mobile devices in the future. As data acquisition becomes more routine, the emergent challenges will fall to issues of data veracity, storage, and integrity, all towards building databases that will serve biosurveillance interests in the future. As has happened with the acquisition and storage of image data, cloud-based repositories for these emerging perspectives on our natural world, akin to tools like iNaturalist (https://www.inaturalist.org/; accessed on 16 April 2021) or Find-A-Pest (http://www.findapest.nz/; accessed on 16 April 2021), will be vital.

## 8. Concluding Remarks

There is much promise emerging around the targeting and capture of NAs in biosecurity monitoring. Technological advances make the capture, extraction, and detection of even early incursions increasingly realistic at time scales where management actions can make a difference to the establishment and spread of NIS. However, standardization and QA/QC protocols for eDNA/eRNA applications are crucial for the biosecurity sector. Data on species presence/absence, which is not properly caveated with an understanding of the limitations of the technology, particularly around its accuracy and precision, may trigger unwarranted management responses [35], cause widespread reluctance of the uptake of the technology by stakeholders, or contribute to the ongoing propagation of unwanted pests.

The nature of the type of data generated is inherently complicated, which means the research community must work closely with biosecurity surveillance managers to ensure protocol steps are feasible, and that data-associated caveats are clearly understood and communicated. This robust approach is particularly important in New Zealand, where the current biosecurity strategy calls for building a team of 5 million (New Zealand's current population) to aid in protecting the country from the spread of unwanted organisms (https://www.mpi.govt.nz/protection-and-response/biosecurity/biosecurity-2025/biosecurity-2025/; accessed on 16 April 2021). In support of these efforts, we envision a (semi)autonomous pipeline that adheres to and expands on minimum reporting guidelines and sampling conduct recommendations, as outlined in Goldberg et al. [129], while overcoming the variability and compromises inherent to environments with a scarcity of clean laboratory resources and experienced personnel. Coupled with systematic efforts and flexibility, a transparent level of standardization between researchers and end-users can be achieved that meets the over-arching goal of protecting human health, economic interests, and environmental resources within the biosecurity sector.

**Author Contributions:** Conceptualization, H.A.B. and A.Z.; funding acquisition, A.Z.; content curation, H.A.B. and A.Z.; writing, H.A.B., A.Z., J.-A.L.S., N.G, X.P., C.D.H.S., U.v.A. and G.-J.J.; review, H.A.B., A.Z., N.G, X.P., C.D.H.S., U.v.A. and G.-J.J.; editing, H.A.B., A.Z., J.-A.L.S., N.G., X.P., C.D.H.S., U.v.A. and G.-J.J. All authors have read and agreed to the published version of the manuscript.

**Funding:** This research was supported by the New Zealand Ministry of Business, Innovation and Employment funding (CAWX1904—a toolbox to underpin and enable tomorrow's marine biosecurity system).

**Institutional Review Board Statement:** Not Applicable.

**Informed Consent Statement:** Not Applicable.

**Data Availability Statement:** No new data were created or analyzed in this study. Data sharing is not applicable to this article.

**Acknowledgments:** Special thanks to Margaret Ryan (Anatomy Department, Otago University) for preliminary data and suggested literature. We also thank two anonymous reviewers for their suggestions that ultimately strengthened the manuscript.

**Conflicts of Interest:** The authors declare no conflict of interest.

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
