# Peer review of "Towards the Optimization of eDNA/eRNA Sampling Technologies for Marine Biosecurity Surveillance"

_water, doi:10.3390/w13081113_

Round 1

Reviewer 2 Report

I have reviewed a review manuscript by Bowers et al. entitled "Towards optimization of eDNA/eRNA sampling technologies for marine biosecurity surveillance." Authors reviewed technologies used in environmental nucleic acids (eNA; eDNA + eRNA) analysis, at the steps of sample collection, post-sampling capture and concentration, preservation, and extraction. The topics addressed in this manuscript are important and in line with current research trends. However, I have major concerns as below: First, I think that the authors should review more of the previous literature. To date, more than 500 papers on environmental DNA analysis of macroorganisms have been published. From my reading of the manuscript, the authors could have cited more previous studies, but they did not. Also, there are many sentences that make claims based on inferences without scientific evidence. For example, authors claim that the use of chemicals including isopropanol, ethanol, and benzalkonium chloride is inadequate for capturing species richness (Lines 311-314), but this statement is not supported by literature. Such inappropriate expressions can be found throughout the manuscript. The authors must make every effort to provide a scientific basis for each claim. Second, This manuscript is a review of eNA, but it contains little information on eRNA. this is unavoidable because there are few examples of actual research on eRNAs in macroorganisms, but eRNAs should be more properly reviewed, for example, by using examples of research on microorganisms. Alternatively, it may be better to focus on eDNA. For these reasons, this manuscript will require significant revision before it is published.
